# Starve Cancer Cells of Glutamine: Break the Spell or Make a Hungry Monster?

**DOI:** 10.3390/cancers11060804

**Published:** 2019-06-11

**Authors:** Jie Jiang, Sankalp Srivastava, Ji Zhang

**Affiliations:** 1Herman B Wells Center for Pediatric Research, Indiana University School of Medicine, Indianapolis, IN 46202, USA; jj15@iu.edu (J.J.); sasriv@iu.edu (S.S.); 2Department of Biochemistry and Molecular Biology, Indiana University School of Medicine, Indianapolis, IN 46202, USA

**Keywords:** glutamine, biosynthesis, amino acid starvation, adaptation, asparagine, aspartate, arginine, glutaminase

## Abstract

Distinct from normal differentiated tissues, cancer cells reprogram nutrient uptake and utilization to accommodate their elevated demands for biosynthesis and energy production. A hallmark of these types of reprogramming is the increased utilization of, and dependency on glutamine, a nonessential amino acid, for cancer cell growth and survival. It is well-accepted that glutamine is a versatile biosynthetic substrate in cancer cells beyond its role as a proteinogenic amino acid. In addition, accumulating evidence suggests that glutamine metabolism is regulated by many factors, including tumor origin, oncogene/tumor suppressor status, epigenetic alternations and tumor microenvironment. However, despite the emerging understanding of why cancer cells depend on glutamine for growth and survival, the contribution of glutamine metabolism to tumor progression under physiological conditions is still under investigation, partially because the level of glutamine in the tumor environment is often found low. Since targeting glutamine acquisition and utilization has been proposed to be a new therapeutic strategy in cancer, it is central to understand how tumor cells respond and adapt to glutamine starvation for optimized therapeutic intervention. In this review, we first summarize the diverse usage of glutamine to support cancer cell growth and survival, and then focus our discussion on the influence of other nutrients on cancer cell adaptation to glutamine starvation as well as its implication in cancer therapy.

## 1. Introduction

More than half a century ago, Harry Eagle noticed that the supplementation of glutamine at millimolar levels in tissue-culturing medium could robustly enhance cell growth and proliferation [1]. Since then, glutamine has become an indispensable nutrient in most modern tissue-culturing media. The reason underlying the usage of exogenous glutamine, which is usually 5- to 20-fold higher than any other individual amino acid in tissue-culturing media, has only recently come into focus is due to enormous progress in the field of cancer cell metabolism [2,3,4]. It is now well-appreciated that glutamine is a versatile biosynthetic substrate to supply carbon and nitrogen atoms for the generation of important precursors for macromolecule biosynthesis [5]. These important precursors include nucleotides, nonessential amino acids (NEAAs) and fatty acids, which are essential building blocks for nucleic acids, proteins and lipids respectively. In addition, glutamine or glutamine-derived metabolites can regulate energy production, redox control, gene transcription and intracellular signaling [6,7]. Thus, targeting glutamine metabolism has shown therapeutic potential in pre-clinical settings through disrupting these growth-promoting processes [2]. 

In contrast to in vitro tissue culture conditions, where glutamine is supplied at millimolar levels, the level of glutamine in tumor tissues in vivo was found significantly lower when compared to the normal surrounding tissues or plasma [8,9,10]. It was thought that increased local consumption of glutamine to support tumor cell growth and poor vascular supply contributed to the observations above mentioned [11]. Although these pioneer studies provided only a snapshot of nutrient status in tumor environment, it set the tone for understanding tumor cells’ responses to glutamine limitation. Furthermore, in certain tumor models, the inhibition of glutamine catabolism failed to provide a therapeutic benefit [12]. Thus, further exploration of the molecular mechanism that tumor cells use to adapt to glutamine limitation will open up avenues for understanding tumor progression in a glutamine-limiting environment, or tumor cells’ responses to therapeutics that disrupt glutamine acquisition and/or utilization. Over the past five years, emerging evidence has suggested that accessibility to other nutrients can influence tumor cells’ dependency on exogenous glutamine, posing the necessity of defining the most limiting metabolite for tumor progression during glutamine starvation. In this review, we will discuss the influences of other NEAAs, including asparagine, aspartate, arginine and cystine, on tumor cells’ dependency on glutamine and their biological/therapeutic implication. 

## 2. Glutamine, a Versatile Biosynthetic Substrate

Glutamine is a NEAA that can be synthesized de novo by using glucose-derived carbon and free ammonium in mammals. Thus, glutamine acquisition through diet is not necessary. Even still, glutamine is one of the most abundant amino acids in human plasma (0.5~0.8 mM), consistent with its versatile usage as a biosynthetic substrate. First, glutamine provides carbon and nitrogen atoms to synthesize nucleotides and other NEAAs. Of note, the carbon and nitrogen atoms can come directly from glutamine or from glutamine-derived metabolites at multiple biosynthetic steps (Figure 1A). For example, during the synthesis of inosine monophosphate (IMP), the precursor of purine, two nitrogen atoms are derived from the γ position of two molecules of glutamine. The third nitrogen is acquired through aspartate. However, this nitrogen atom in aspartate is indeed derived from glutamate through transamination; while glutamate can be produced from glutamine by several reactions that remove the nitrogen atom from the γ position of glutamine. Thus, the third nitrogen in IMP is mainly derived from the α position of glutamine. Similarly, during the synthesis of uridine monophosphate (UMP), the precursor of pyrimidine, glutamine contributes both the γ and α nitrogen atoms. However, the incorporation of α nitrogen from aspartate is accompanied by three carbon atoms of aspartate to build the orotate ring. Since the carbon backbone of aspartate is derived from the tricarboxylic acid cycle (TCA) cycle, which is replenished by glutamine-derived α-ketoglutarate, glutamine is also the major source of carbon atoms for UMP (Figure 1A). 

Second, glutamine can drive the TCA cycle and ATP production. One reason that cancer cells rely on high levels of exogenous glutamine is because glutamine can be used to fuel the TCA cycle through α-ketoglutarate to allow its further oxidation [13]. It was shown that glutamine depletion reduces the NADH/NAD^+^ ratio, which inhibits oxygen consumption and ATP production [14]. Thus, in addition to replenishment of the TCA cycle intermediates for biosynthesis, continuous oxidation of glutamine-derived α-ketoglutarate through the TCA cycle also provides a source of energy (Figure 1B). 

Third, glutamine supports glutathione biosynthesis and NADPH production to mitigate oxidative stress. Glutamine supports the biosynthesis of glutathione, a primary cellular antioxidant, through at least two mechanisms (Figure 1B). Since de novo biosynthesis of glutathione requires glutamate, cysteine and glycine, glutamine-derived glutamate is thus a direct substrate of glutathione biosynthesis. In addition, intracellular cysteine is mainly obtained through reduction of cystine, which is imported into cells at the expense of glutamate exportation through the xCT transporter. It is for the same reason that xCT-positive triple negative breast cancer cells are sensitive to glutamine starvation due to a failure to maintain intracellular cysteine for glutathione production and anti-oxidative defense [15]. Furthermore, glutamine can contribute to redox balance via NADPH production. However, in this scenario, glutamine does not function as a substrate. Rather it involves glutamine catabolism through the TCA cycle to produce aspartate, which is shuttled into the cytosol and subsequently converted to oxaloacetate, malate and pyruvate [16]. The final step of these reactions is catalyzed by malic enzyme 1 (ME1) that passes electrons to NADP^+^ to generate NADPH (Figure 1B). Along this line, glutamine-dependent redox control creates additional vulnerability in KRAS-driven lung cancers containing KEAP1 mutations, which activates NRF2 pathway to mitigate the oxidative stress [17,18].

## 3. Non-Biosynthetic Role of Glutamine

The fact that glutamine has become a central player of amino acid metabolism in cancer should also be attributed to its non-biosynthetic functions. It has been shown that glutamine-derived metabolites not only provide precursors for macromolecules, but also function as substrates or co-factors to regulate cellular processes. A compelling example is the glutamine-derived α-ketoglutarate that serves as a co-substrate for a set of dioxygenase enzymes to mediate DNA and histone demethylation (Figure 1B). It was reported that glutamine deficiency in the central core of tumor tissues contributed to histone hypermethylation in a model of melanoma through an α-ketoglutarate-dependent mechanism [23]. As a result of global histone hypermethylation, the gene expression was altered, which led to dedifferentiation and resistance to a BRAF inhibitor in a xenograft mouse model. We anticipate that further exploration with other tumor models hijacking epigenetic machinery for their progression will provide a broader understanding of the impact of glutamine availability on gene expression during tumor progression or response to chemo-agents. Of note, glutamine-derived α-ketoglutarate in gene expression control through histone and DNA demethylation has also been demonstrated in murine embryonic stem (ES) cells, T lymphocytes and macrophages [24,25,26,27,28]. Deeper understanding of the signal component that regulates glutamine accessibility or catabolism to convey global gene expression control in both tumor cells and immune cells will open new opportunities to unravel the complexity of the tumor environment to facilitate development of more effective immunotherapies. 

Glutamine is also a positive regulator of growth-promoting signaling. The best characterized signaling activated by glutamine is the mammalian target of rapamycin complex 1 (mTORC1) (Figure 1B). It was first reported that intracellular glutamine can be used as a counter ion to exchange for extracellular essential amino acids (EAAs), which are critical for mTORC1 activation and mTORC1-dependent cell growth [29]. Later on, biochemical analyses done in mammalian cells and yeast showed that glutamine can directly activate mTORC1, but independently of Rag GTPase that is required for leucine-dependent mTORC1 activation [30,31]. In addition, glutamine-derived α-ketoglutarate is involved in mTORC1 recruitment to the lysosome for its activation [32]. Of interest, a recent report showed that glutamine starvation can inhibit MYC protein accumulation and MYC-dependent gene transcription in colorectal cancer [33] (Figure 1B). Although the mechanism is yet to be elucidated, it involves glutamine-dependent adenosine production and translation control through the 3′ UTR of MYC mRNA. Since MYC-dependent reprogramming of glutamine metabolism is well established [34,35], this work illustrates a reciprocal regulation between MYC and glutamine metabolism, which can profoundly impact the understanding of the interplay between nutrient and oncogenic signals. 

## 4. Glutamine Starvation: An Experimental Condition or Pathophysiological Stress?

Themes of glutamine starvation were first brought to attention by nutritionists who noticed that plasma glutamine levels reduce significantly during severe injury [36]. Similar phenomena were observed later on with many other pathological conditions, which were often associated with increased catabolic activities to compensate the loss of circulating glutamine. For example, muscle protein catabolism is used to maintain plasma glutamine concentration at a lower level during long-term fasting [37]. In cancer patients, plasma glutamine concentration is also reduced [38]. It was thought that tumor-associated complications, such as cachexia, may be a consequence of reduced concentrations of amino acids in the circulation. As a result, the supplementation of glutamine in the diet was found to reduce muscle breakdown and enhance immune function in certain cancer patients receiving chemotherapy [39]. 

Considering the fact that cancer cells increase glutamine consumption to support biosynthesis, the observed reduction of glutamine concentration in the blood stream of cancer patients raises the belief that glutamine limitation may occur during tumor progression. Indeed, glutamine levels in tumor tissues or body fluids around tumor tissues were found lower than normal tissues or plasma [8,9,10,40]. Furthermore, recent work suggests that even within the same tumor tissue, glutamine was found further depleted in the core of xenograft tumors when compared to the periphery of the tumors [23]. This result is consistent with the concept that inadequate vascular supply of nutrient remains a barrier for inner tumor mass accumulation [41]. Thus, extensive efforts have been made recently to understand various mechanisms that tumor cells use to adapt to glutamine limitation for their further progression. These mechanisms include signaling-mediated cell fate decision, proteolytic scavenging, enhanced de novo biosynthesis of glutamine and rewiring the utilization of other nutrients [5]. 

## 5. Influence of Other Amino Acids on Glutamine Starvation

Of interest, studies from the past five years suggest that accessibility to other nutrients, particularly NEAAs, can profoundly affect tumor cells’ responses to glutamine starvation [42,43,44,45,46,47]. Asparagine, aspartate and arginine demonstrate the capacity to protect tumor cells from glutamine starvation or from block of glutamine catabolism; while cystine uptake increases cellular sensitivity to these perturbations. In the next few sections, we will discuss the impact of each of these findings on understanding glutamine metabolism as well as their potential therapeutic applications in cancer. 

### 5.1. Asparagine

Asparagine, a NEAA that can be synthesized de novo from glutamine, was found to be able to suppress glutamine-depletion-induced apoptosis in brain tumor cells [42]. Since all tested brain tumor cells were maintained in standard DMEM medium that contains 6 mM glutamine but without five NEAAs, including alanine, proline, glutamate, aspartate and asparagine, the authors performed comprehensive metabolomic analysis to quantify intracellular metabolites following glutamine depletion in the presence or absence of extracellular supplementation of asparagine. Of interest, asparagine supplementation does neither rescue the levels of other four NEAAs (Ala, Pro, Glu and Asp) nor any TCA cycle intermediate, both of which are significantly reduced by glutamine depletion [42]. This work highlights the importance of other nutrients in modulating tumor cells’ responses to glutamine starvation and also brings an important question of what is the most critical glutamine-derived metabolite for tumor cells to survive glutamine starvation. Mechanistically, asparagine rescues cell survival at least partially through inhibiting endoplasmic reticulum (ER) stress that is caused by glutamine starvation [42]. Later on, the same group showed that the effect of asparagine to support cellular adaptation to glutamine starvation is generalizable in multiple tumor types [44]. In several epithelial breast tumor lines, asparagine was shown to even rescue the proliferation defect during glutamine starvation without restoring other NEAAs and TCA cycle intermediates at steady levels [44]. In this work, the authors showed that asparagine can rescue global protein synthesis that is suppressed by glutamine starvation, and this process is facilitated by an enhanced expression of glutamine synthetase (GLUL) at post-transcriptional level in an asparagine-dependent manner [44]. 

These two works indicate that most glutamine-dependent biosynthetic activities can still proceed, with the exception of asparagine production, when exogenous glutamine is absent. Indeed, work done in vasculature-forming endothelial cells and Kaposi’s sarcoma-associated herpesvirus (KSHV)-transformed fibroblast cells showed a similar effect of asparagine to support cell survival and proliferation following glutamine depletion [48,49]. Importantly, asparagine can rescue glutamine-depletion-induced growth defect in breast tumor cells at concentrations within physiological range (25~100 μM) [44]. At this range of concentrations, none of the other NEAAs can rescue proliferation. Furthermore, introduction of a catalytically active asparaginase, an enzyme to break down asparagine, from lower vertebrates into mammalian cells prevents the ability of asparagine to rescue glutamine starvation at physiological concentrations. Since the putative human asparaginase shows minimal asparaginase activity [44], the authors speculate an evolutionary pressure to cause the selective loss of its activity in mammalian cells as a means to preserve intracellular asparagine in order to mediate cellular adaptation to pathophysiological changes of environmental glutamine levels [44]. 

Furthermore, like glutamine, intracellular asparagine was also found to be able to function as a counter ion for the exchange of extracellular EAAs to sustain mTORC1 activity [29,50]. Supplementation of asparagine in DMEM medium that does not routinely contain asparagine can sustain mTORC1 activity, which is otherwise suppressed following glutamine deprivation [44]. Whether restoration of mTORC1 activity by asparagine supplementation plays a role in tumor cell growth/survival during glutamine starvation remains to be determined. 

### 5.2. Aspartate and Arginine

In addition to asparagine, two recent papers have shown that aspartate, another NEAA that can be synthesized de novo from glutamine-derived carbon and nitrogen atoms, plays a critical role in tumor cells’ adaptation to glutamine starvation or the inhibition of glutamine catabolism [43,45]. Tajan et al. (2018) showed that aspartate uptake through its cell surface transporter SLC1A3 can support tumor cell proliferation under glutamine starvation [45]. In this work, p53 tumor suppressor is required for the transcriptional induction of SLC1A3 following glutamine depletion. In another independent work, Alkan et al. (2018) showed that release of aspartate from mitochondria into the cytosol plays a critical role to mediate tumor cell survival and proliferation in a low glutamine environment or following pharmacological inhibition of glutaminase, a key enzyme for glutamine catabolism and further utilization [43]. This export of mitochondrial aspartate into the cytosol is mediated by the mitochondrial aspartate-glutamate carrier 1 (AGC1), and genetic inhibition of AGC1 compromises xenograft tumor growth when a glutaminase inhibitor is used. Mechanistically, both studies showed that aspartate is a critical precursor for nucleotide biosynthesis, but dependence on aspartate to support the TCA cycle intermediates and other NEAAs is not required for cell growth during glutaminase inhibition [43,45]. 

Arginine was recently found to be able to support tumor cell adaptation to glutamine starvation [46]. It was shown that glutamine deprivation induces the expression of arginine transporter SLC7A3 on cell surface in a p53-dependent manner to facilitate arginine uptake. Consequently, the intracellular accumulation of arginine sustains mTORC1 activation and cell growth/proliferation. Of interest, arginine does not contribute to the TCA cycle through ornithine to glutamate conversion or fumarate release during the urea cycle [46]. Whether it can support the biosynthesis of other macromolecules during glutamine starvation warrants further investigation. 

### 5.3. Cystine

Unlike asparagine, aspartate and arginine, cystine was found to increase tumor cells’ sensitivity to glutaminase inhibition, when supplemented above physiological levels in tissue-culturing medium [47]. Since cystine uptake through the xCT transporter uses glutamate as a counter ion in exchange, it was proposed that the cystine/glutamate exchange creates dependency on glutaminase to maintain glutamine deamination and glutamate production [47]. Of interest, cystine import has been reported as a central component of glutamine/glutamate-dependent anti-oxidative defense [15]. These works together may suggest that a precise control of glutamine utilization is needed for optimized tumor growth, particularly when exogenous glutamine supply is compromised. Tumor cells rely on glutamine-derived glutamate for the exchange of cystine, which is the major source of intracellular cysteine, a NEAA for protein synthesis and glutathione production; on the other hand, the excessive uptake of cystine may cause glutamate exhaustion, thus preventing its further utilization through the TCA cycle or transamination. 

## 6. What Is the Critical Limiting Metabolite during Glutamine Starvation?

All the results above mentioned suggest the existence of a comprehensive program to mediate tumor cell adaptation to glutamine limitation. However, the fact that more than one single amino acid can alter tumor cells’ responses to glutamine starvation or its catabolic inhibition creates a challenge for the efficacy of targeting glutamine metabolism in cancer. An obvious question one may have is why tumor cells choose different amino acids to modulate their dependencies on exogenous glutamine. The simplest answer is that the intrinsic property of tumor cells differs. This specificity can be dictated by tumor origin, oncogene/tumor suppressor status and tumor microenvironment. For example, one may predict that tumors with loss of function of p53 may not be able to adapt to glutamine starvation through uptake of aspartate or arginine. Further investigation is needed for the development of complementary strategies that should be combined when targeting glutamine metabolism in cancer.

Another way to reconcile the diversity of other amino acids to support tumor cell adaptation to compromised glutamine metabolism is to define the most critical limiting metabolite. The fact that asparagine alone is able to rescue cell proliferation without restoring the TCA cycle intermediates and other glutamine-derived NEAAs following glutamine depletion provides a compelling example [44]. In this work, the authors show that supplementation of exogenous glutamine at 0.1 mM cannot support cell proliferation as well as the same concentration of asparagine. Since cell proliferation requires both glutamine and asparagine, this result suggests that the biosynthesis of asparagine, but not glutamine itself, is the limiting factor for epithelial breast tumor cells to proliferate when exogenous glutamine is low (Figure 2A,B). It is worthy of note that asparagine was also found to be lower in the tumor tissues when compared to surrounding normal tissues [40]. In the core of xenografted tumor tissues asparagine was found to be around 30 μM, which is 25% of the asparagine levels in the periphery of the same tumor tissues [23]. At this concentration, it is likely that asparagine is sufficient to support protein synthesis to allow limited amount of glutamine to be used for other biosynthetic reactions [44].

However, one may question why tumor cells cannot simply use glutamine-derived carbon and nitrogen to synthesize asparagine through asparagine synthetase (ASNS), an enzyme highly expressed in solid tumor cells [51], if glutamine itself is not limiting. A clue may be found from a recent report showing that the relative biosynthetic energy cost for asparagine is the highest among the nine NEAAs (Asn, Asp, Gln, Ser, Arg, Pro, Glu, Ala, Gly) that are synthesized from glycolysis and TCA cycle-derived carbon source in humans [52]. This is likely because of the high energy cost to maintain the carbon flux from glucose to go through the TCA cycle to generate glutamate, glutamine and aspartate when exogenous glutamine is absent. We can speculate that although tumor cells express high levels of ASNS, they are unable to afford the additional energy for asparagine production. Notably, glutamine itself is required for protein synthesis. In this regard, asparagine facilitates glutamine synthesis de novo through a post transcriptional induction of glutamine synthetase (GLUL) during glutamine starvation [44]. 

## 7. Key Variants Impacting the Definition of Critical Limiting Metabolite 

Defining the critical limiting metabolite in each context will not only elucidate the fundamental connection between the biochemistry of glutamine metabolism and its biological functions, but will also provide new metabolic targets that can be inhibited to enhance the efficacy of targeting glutamine metabolism in cancer. However, to define the critical limiting metabolite, several factors need to be considered rigorously. 

First, the amino acid composition in the tissue-culturing medium defines the types of starvation. In the setting of Dulbecco’s modified Eagle’s medium (DMEM), which does not contain exogenous asparagine and aspartate, glutamine deprivation is anticipated to cause intracellular depletion of at least these three amino acids [42,44,46]. In contrast, Medium 199 contains all the NEAAs with the exception of asparagine, and thus glutamine deprivation may cause depletion of glutamine and asparagine, but not aspartate [48]. Similarly, in DMEM supplemented with all the NEAAs, glutamine deprivation may only deplete glutamine itself unless aspartate uptake is blocked [45]. In this case, uptake of excessive aspartate through its cell surface transporter not only can rescue aspartate-dependent nucleotide and protein synthesis, but also can support the replenishment of the TCA cycle intermediates to drive the de novo biosynthesis of glutamine (Figure 2C). 

Second, glutamine deprivation and glutaminase inhibition may define different types of critical limiting metabolite. In experimental conditions where exogenous glutamine is depleted or reduced to a low level, one may predict that all glutamine-dependent cellular processes are compromised. However, glutaminase inhibition is anticipated to block glutamine to glutamate conversion and its further utilization in the TCA cycle, NEAA production and nucleotide biosynthesis. In this setting, intracellular glutamine itself and any glutamine-dependent growth/survival signaling are not expected to be limiting. The fact that exogenous aspartate rescues glutaminase inhibition through nucleotide synthesis but not the TCA cycle and other NEAAs suggests that these metabolic activities can proceed, likely by using other carbon sources in the tissue-culturing medium [43] (Figure 2D). It needs to be noticed that glutamine is also an indispensable nitrogen source for nucleotide biosynthesis independently of aspartate (Figure 1A). The fact that glutamine itself is unlimited following glutaminase inhibition may explain why these cells do not require exogenous aspartate to support the TCA cycle, a necessary process for de novo biosynthesis of glutamine. 

Third, the critical limiting metabolite that is responsible for cell proliferation and survival can be different under glutamine limitation. Since cell proliferation requires biomass accumulation, one may anticipate that when a single amino acid can rescue cell proliferation during glutamine deprivation or glutaminase inhibition, biosynthesis of all the macromolecules must be able to continue (Figure 2B–D). In this setting, tumor cells must retain their capacity to acquire an adequate amount of amino acids, including asparagine, aspartate, glutamate and glutamine itself for protein synthesis. In addition, some of these amino acids must be acquired at levels enough to support the biosynthesis of other macromolecule precursors. For example, aspartate and glutamine are indispensable precursors for nucleotide biosynthesis. However, in the setting where a single amino acid can only rescue the survival defect, such as arginine, macromolecule biosynthesis is not necessary (Figure 2E). Indeed, it still remains elusive what the most critical limiting metabolites are for cell survival under various types of metabolic stress. However, alterations in signaling pathways involving nutrient sensing, stress response and apoptosis are thought to be the central players [42,46,53].

## 8. Complexity of Glutamine Starvation in Tumors In Vivo 

Unlike in vitro cell culture experiments where we can precisely control the amount of nutrient in tissue-culturing media, the nutrient accessibility in the tumor environment fluctuates. In general, tumors with poor vascular supply of nutrient have a higher chance to suffer from glutamine starvation as well as to engage adaptive mechanisms to mitigate the stress. Even with efficient vascular supply, the concentration of glutamine in the blood stream and body fluids is several folds lower when compared to most tissue-culturing media. On the other hand, unlike in vitro experiments where tumor cells are often subjected to extreme glutamine deprivation, the stress caused by glutamine limitation in tumor cells in vivo may be compensated to a certain extent by a low rate of continuous supply. To determine tumor cells’ dependency on exogenous glutamine at a nutritional status close to physiological conditions, Tardito et al. (2015) developed a new formula of medium containing nutrient at concentrations close to the human serum. In this medium, glioma cells can grow without exogenous glutamine [54]. Of interest, the glutamine that is required for cell growth in this setting can be synthesized de novo through GLUL in glioma cells or astrocytes in the tumor environment [54]. 

The ability of glioma cells to acquire glutamine from astrocytes reminds us of the importance of other cell types co-existing in the tumor microenvironment [55]. In models of ovarian cancer and prostate cancer, it was shown that cancer-associated fibroblasts (CAFs) can synthesize glutamine de novo and secrete glutamine to support tumor cell growth in a glutamine-limiting environment [56,57]. In addition to stromal cells, macrophages in the tumor environment can contribute to tumor progression through glutamine synthesis. It was shown that macrophage polarization during glutamine starvation favors M2-like fate in a GLUL-dependent manner, which facilitates immunosuppression and tumor metastasis [58]. Blocking GLUL in macrophages restored the M1-like phenotype, promoted cytotoxic T cell function and inhibited metastasis. However, whether macrophages depend on glutamine biosynthesis to support their own macromolecule biosynthesis or glutamine-dependent signaling and gene expression warrants further investigation. The stroma-dependent nutrient supply is not limited to glutamine. In a model of prostate cancer, stroma cells can synthesize and secrete asparagine for tumor cells to adapt to a glutamine-limiting environment [59]. This ability to synthesize asparagine requires activating transcription factor 4 (ATF4)-dependent expression of ASNS in stroma cells. However, one may speculate that stroma cells use a different mechanism than tumor cells for glutamine acquisition, as de novo synthesis of asparagine in the absence of exogenous glutamine is not energetically favorable [44,52]. Furthermore, a recent work shows that glutamate/aspartate exchange between tumor cells and stromal fibroblast cells is important for maintaining tumor cell growth and stroma cell function [60]. Whether this unique dependency on glutamate/aspartate exchange reflects a limitation of environmental glutamine remains to be elucidated. 

## 9. Therapeutic Implication

Since the first description of “glutamine addiction” in cancer cells, the potential to target glutamine metabolism has been extensively studied in pre-clinical models. However, the existence of various adaptive mechanisms to mitigate the stress caused by glutamine limitation plus a complicated tumor microenvironment can create practical challenges. Due to toxicity and the lack of specificity, most chemical inhibitors of glutamine uptake and catabolism are still used as tool compounds [2]. The best developed small molecule is CB-839, a potent inhibitor of the mitochondrial glutaminase (GLS) and the only one that is currently used in clinical trials in cancer patients [61]. Since GLS catalyzes the conversion of glutamine to glutamate, it is anticipated that the treatment will deplete intracellular glutamate and block its further utilization in the TCA cycle, NEAA production and nucleotide biosynthesis. However, as discussed in Section 7, the physiological responses to glutaminase inhibition are likely to be different from glutamine deprivation in tumor cells. In mouse models of lung cancer and glioma, glutamine-derived carbon entry into the TCA cycle is limited in vivo, which correlates with resistance to CB-839 treatment [12,62]. These results do not exclude the importance of glutamine to support tumor growth, but rather suggest that these tumors do not rely on glutamine-derived carbon to support the TCA cycle. In this regard, the importance of glutamine-derived nitrogen for tumor growth in vivo has not been evaluated, partly because most of these reactions do not rely on glutaminase activity. Further exploration is necessary to determine whether the limited dependence of certain tumors on glutamine-derived carbon to support the TCA cycle is a choice that tumor cells make or simply a consequence of limited glutamine supply. In either case, can we identify unique features in tumor tissue of origin or oncogene/tumor suppressor status that are associated with their dependency on glutaminase? As mentioned previously, the availability of glutamine in the tumor environment is often limited [8,9,10]. The ability of tumor cells to use both glutamine-derived carbon and other carbon sources to fuel the TCA cycle may reflect the metabolic plasticity based on environmental nutrient accessibility. 

Another difference between glutaminase inhibition and glutamine deprivation is the existence of other glutamine to glutamate converting enzymes that can compensate when glutaminase is inhibited. These include the enzymes in the biosynthesis of asparagine, nucleotides, NAD and glucosamine [5]. All these enzymes transfer the γ nitrogen of glutamine to their substrates and thereby generating glutamate as a byproduct. It is still unclear whether these pathways can compensate in certain cases where glutaminase inhibition is not effective. To prevent glutamine acquisition for therapeutic intervention, which mimics glutamine deprivation, a recent work identified a small molecule inhibitor, V-9302 that blocks the function of glutamine transporter (ASCT2) [63]. In this work, V-9302 blocked glutamine uptake in a broad spectrum of solid tumor lines as well as several xenograft tumor models, which led to a profound defect in tumor cell growth and survival. However, ASCT2 is also used for the import of other amino acids, such as leucine, and V-9302 can also block leucine uptake [63]. It will be important to determine the potential toxicity in normal cells that may use ASCT2 to import amino acids. Furthermore, exploration of ASCT2-dependency in different types or subtypes of tumor is necessary to direct the application of this new compound [64]. In the future, exploring glutamine dependency in additional tumor models with more sophisticated considerations at different stages of tumor progression, including initiation, local infiltration, metastasis or response to chemotherapy, will provide further insights into the therapeutic impact of understanding glutamine metabolism in cancer at physiological conditions. 

Since tumor cells can use other amino acids to mitigate the stress caused by glutamine limitation or inhibition of glutamine catabolism, perturbing the acquisition and metabolism of these amino acids should be considered at least as a complementary approach when we target glutamine metabolism in cancer. For example, exogenous asparagine is a limiting nutrient for acute lymphoblastic leukemia (ALL) cells to grow due to their inability to synthesize asparagine de novo [51]. As a result, L-asparaginase, a bacterial enzyme that can degrade asparagine in the blood stream, has been used for decades to successfully treat pediatric ALL patients [65,66]. However, L-asparaginase is not effective in treating solid tumors, likely due to their capacity to synthesize enough asparagine. But combining L-asparaginase with inhibitors of glutamine metabolism may be a reasonable strategy to enhance the therapeutic outcome. Of interest, a recent work showed that L-asparaginase treatment alone is able to reduce the chance of breast cancer metastasis to lung without altering tumor growth at primary sites [67]. Whether this result reflects a limitation of glutamine in lung or during the process of metastasis warrants further investigation. Similar to L-asparaginase, arginine-depleting enzyme, arginine deiminase (ADI), is being used in multiple clinical trials for solid tumors based on the observation that certain types of tumor lack the expression of arginine biosynthetic enzymes and are thus arginine auxotrophic [68]. Whether ADI can be combined with glutamine metabolic inhibition for optimized therapeutic outcome is worthy of investigation. 

## 10. Conclusions

Despite an enormously growing understanding of glutamine metabolism and its diverse biological functions in cancer, challenges remain before this pathway can be broadly targeted to treat cancer patients. First, we need to define specific mechanisms mediating tumor cell adaptation to glutamine limitation. Tumor cells can adapt to glutamine limitation through various mechanisms, reflecting a metabolic plasticity that may be selected during tumor evolution as a means of optimized nutrient utilization. The specific mechanism that a tumor cell chooses is dictated by tumor type or subtype, oncogene/tumor suppressor status, tumor site and stages of tumor development. Defining a specific adaptive mechanism can not only facilitate the decision of targeting glutamine metabolism or not, but also provide insights into new pathways that may need to be inhibited together to maximize the efficacy of targeting glutamine metabolism. Second, since the existence of other nutrients can profoundly affect tumor cell response to glutamine limitation, it is critical to define the precise nutrient status in the tumor environment. Currently, most mass spectrometry (MS)-based quantification of nutrients from primary tumor tissues do not distinguish between nutrient levels inside tumor cells or in the surrounding tumor environment. Furthermore, an efficient method needs to be developed to timely separate tumor cells, stromal cells and immune cells from the same tumor tissue for metabolite measurement. Third, we need to further understand the metabolic interaction between tumor cells and immune cells, such as T cells, when glutamine uptake or catabolism is compromised. It has come to light that many glutamine-dependent cellular functions have gone beyond cancer cells. In T cells, perturbation of glutamine acquisition and catabolism has a profound effect on T cell differentiation and immune response [25,69,70]. Thus, dissecting differential responses to the limitation of glutamine or glutamine-derived metabolites between tumor cells and T cells will ensure therapeutic strategies that destroy tumor growth while preserve effector and cytotoxic T cell function. In the future, it is our anticipation that evidence will show up to fill in the gaps in our understanding of these areas. At that time, we would no longer need to worry about creating a “hungry monster” by starving cancer cells of glutamine.

## Figures and Tables

**Figure 1 cancers-11-00804-f001:**
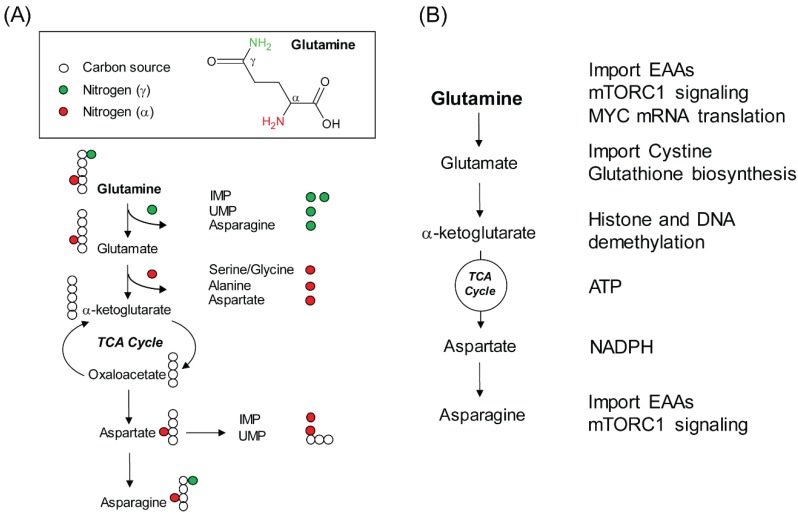
Glutamine catabolism and its cellular functions. (**A**) Utilization of glutamine-derived carbon and nitrogen atoms for the synthesis of nucleotides and NEAAs. The chemical structure of glutamine is illustrated at the top. This diagram highlights the contribution of carbon and nitrogen atoms from glutamine or its catabolic products to nucleotides and NEAAs. The biosynthesis of proline and arginine can also be derived from glutamate, but is not illustrated here due to the space limitation. Both amino acids use the entire five carbon atoms and one nitrogen atom of glutamate as their backbones. The biosynthesis of arginine acquires three additional nitrogen atoms. One is derived from glutamate, another one is derived from aspartate and the last one is derived from free ammonium. However, de novo biosynthesis of arginine is not considered to be the standard route of arginine acquisition, because several key enzymes in the urea cycle, which is indispensable for arginine production, are silenced in most cancers [19,20,21,22]. As a result, arginine is thought to be a conditionally essential amino acid. In addition, the γ-nitrogen of glutamine is also used for the de novo synthesis of GMP from XMP, CTP from UTP, NAD and glucosamine, which are not illustrated in the diagram. (**B**) Cellular functions of glutamine and glutamine-derived metabolites beyond a donor of carbon and nitrogen atoms for the precursors of macromolecules.

**Figure 2 cancers-11-00804-f002:**
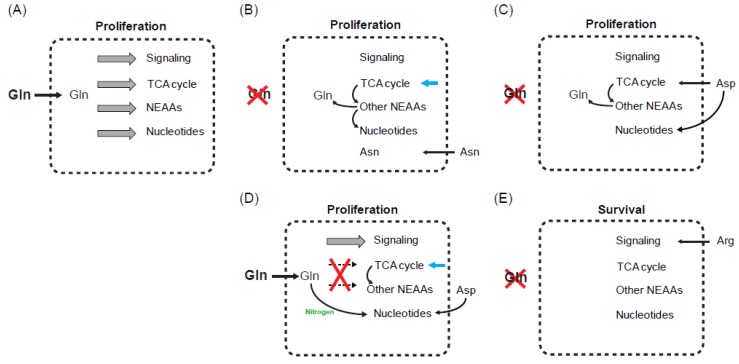
Possible paradigms of defining the critical limiting metabolite. (**A**) In glutamine-replete condition, glutamine is used for many biosynthetic and non-biosynthetic purposes (grey boxed arrow), which together contribute to tumor cell proliferation. For simplification, we listed signaling, TCA cycle, NEAAs and nucleotides as key components mediating glutamine-dependent cell proliferation. In addition, continuation of some components may rely on the other components as we discussed. (**B**) Following glutamine deprivation, all the glutamine-dependent signaling, TCA cycle activity, and NEAA and nucleotide biosynthesis are reduced. Supplementation of asparagine rescues proliferation without restoring other NEAAs and the TCA cycle intermediates [44]. Cells must use other carbon sources (blue arrow) to support the TCA cycle and the biosynthesis of NEAAs including glutamine. (**C**) Exogenous aspartate can rescue cell proliferation following glutamine deprivation [45]. In this situation, aspartate functions as a biosynthetic substrate to replenish the TCA cycle and synthesize other NEAAs and nucleotides. It remains to be determined whether aspartate directly contributes to growth promoting signal or not. (**D**) Glutaminase inhibition may only block glutamate production and its subsequent usage in the TCA cycle and NEAA/nucleotide synthesis. In this setting, glutamine itself or glutamine-dependent growth promoting signaling is not affected. Aspartate was shown to be able to rescue proliferation through supporting nucleotide biosynthesis, but not the TCA cycle and other NEAAs [43]. Similar to (**B**), cells must use other carbon sources (blue arrow) to support the TCA cycle and other NEAAs. (**E**) In this situation, sustaining a survival signaling is required for tumor cells to escape from glutamine-deprivation-induced apoptosis. Thus, it usually does not require the restoration of the TCA cycle and the biosynthesis of NEAAs and nucleotides.

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
