# Peer review of "Starve Cancer Cells of Glutamine: Break the Spell or Make a Hungry Monster?"

_cancers, 2019, doi:10.3390/cancers11060804_

Round 1

Reviewer 1 Report

The authors present a well-written and beautifully illustrated review on the role of glutamine and in cancers with a specific focus on metabolic adaptations to glutamine starvation. A well-reasoned part of this review is detailed discussion on the role played by other amino acids including asparagine, aspartate, arginine and cystine on glutamine starvation in the tumor microenvironment. The authors also cover therapeutic potential of inhibiting glutamine metabolism and possible pathways of resistance development. Overall, this review article is thorough, discusses several aspects of glutamine metabolism and will provide a good resource to researchers in the field.

Author Response

We thank the reviewer for recommending our manuscript for publication.

Reviewer 2 Report

In this review, Jiang and colleagues provide a comprehensive overview of the role of glutamine metabolism in cancer cells. Targeting glutamine metabolism for cancer therapy has long been a goal of the cancer metabolism community, but current efforts to translate glutamine-targeted therapies from in vitro to in vivo have identified numerous challenges. This review discusses these challenges and highlights mechanisms by which cells can adapt to glutamine starvation. Overall this is a very thorough discussion of this topic that contains many references that will be useful for readers. Below are a couple of suggestions that might help to improve this very nice review. 

The authors very briefly mention that depletion of glutamine and inhibition of glutaminase may have significantly different effects on cancer cell physiology. This is an important point is glutaminase inhibitors are the primary mechanism of inhibiting glutamine metabolism that is currently being explored clinically. It might help if the authors expand their explanation of the differences between glutamine deprivation and glutaminase inhibition so that readers are better able to distinguish what physiological responses to glutamine deprivation are likely relevant to glutaminase inhibition as well.

At times the there are issues with the written english that make the review somewhat difficult to read. A quick proofreading would significantly improve the readability of this review. 

Author Response

We thank for the reviewer’s agreement that our manuscript provides a thorough discussion on glutamine metabolism and tumor cell adaptation to glutamine starvation. We agree with the reviewer that glutamine starvation and glutaminase inhibition may have different effects on tumor cell response. Therefore, we have expanded our discussion in Section 9 in light of the fact that glutaminase inhibitors are the primary focus in clinical settings. Two major issues with glutaminase inhibition are: (1) it does not block the accumulation of glutamine in cells and thus can not prevent the usage of the γ-nitrogen of glutamine; (2) other γ-nitrogen consuming reactions can potentially bypass the requirement of glutaminase to generate glutamate from glutamine. We believe that the revised manuscript will now help readers understand what physiological responses to glutamine starvation are likely relevant to glutaminase inhibition. We also did proofreading of our written English and hope it has improved the readability of this review.

Reviewer 3 Report

Jiang and colleagues describe an elegant and comprehensive review of the critical role of glutamine metabolism in various cancers. The authors discuss glutamine as a nutrient for biosynthetic processes, glutamine as a signaling molecule for mTORC1 activity, glutamine as a key player in reducing power and finally the potential implication of glutamine and other amino acids in cancer therapy. The authors provide an interesting discussion and paragraph about the limiting metabolite for tumor growth.

·         The authors mentioned that glutamine concentration is generally low in the tumor microenvironment, it could be remarkable if the authors can also report the actual surrounding levels of asparagine in the microenvironment, given the critical role of this metabolite for tumor growth.

·         The authors could discuss this recent paper on the adaptive benefit provided by the activation of the KEAP1/NRF2 pathway in KRAS/LKB1-driven tumors that enhances glutamine addiction (PMID: 31040157).

·         Fig. 1A needs to be simplified. On the left side of panel 1A, the drawn carbon and nitrogen rings are slightly confusing and need to be amended.

Author Response

We thank for the reviewer’s comments and suggestions on our manuscript. We have included a brief discussion on asparagine levels in tumor environment in Section 6, which are likely sufficient to mediate tumor cell adaptation to glutamine limitation. Furthermore, we have included the discussion of KEAP1/NRF2 pathway in glutamine addiction through regulating the anti-oxidative defense in Section 2. Finally, the panel (A) of Figure 1 has been simplified. We removed proline and arginine from the diagram and included additional arrows to emphasize the nitrogen flux. We hope that the revised Figure 1(A) can highlight the transfer of nitrogen atoms from the a and g positions of glutamine to other substrates and the carbon usage through the TCA cycle. However, we did include a brief description of proline and arginine biosynthesis in the figure legend.